# Quality of Life in Women over 65 Years of Age Diagnosed with Osteoporosis

**DOI:** 10.3390/ijerph19095745

**Published:** 2022-05-09

**Authors:** María Belén Pastor-Robles, Agustín Mayo-Íscar, Rosa M. Cárdaba-García, Virtudes Niño-Martín

**Affiliations:** 1Castilla y León Blood Therapy and Donation Center, 47007 Valladolid, Spain; 2Department of Statistics and Operational, Research and Mathematics Institute, University of Valladolid (IMUVA), 47007 Valladolid, Spain; agustinm@eio.uva.es; 3Nursing Department, Faculty of Nursing, University of Valladolid, 47005 Valladolid, Spain; rosacardaba@yahoo.es (R.M.C.-G.); vninoger@gmail.com (V.N.-M.); 4Nursing Care Research (GICE—Grupo de Investigación de Cuidados de Enfermería), Faculty of Nursing, University of Valladolid, 47005 Valladolid, Spain; 5Emergencies Management (SACYL—Sanidad Castilla y León), 40002 Segovia, Spain; 6Primary Care Management Valladolid East (SACYL), 47010 Valladolid, Spain

**Keywords:** hip fracture, menopause, osteoporosis, quality of life, risk factors

## Abstract

(1) Background: Today’s society is moving towards active aging, underlining the importance of understanding and improving quality of life (QoL). This QoL in women over the age of 65 years diagnosed with osteoporosis was compared with the QoL of the general population, and risk factors for osteoporosis related to QoL were identified. (2) Methods: This is an observational, descriptive, cross-sectional study with a personal interview. The study population was 704 women over 65 years of age diagnosed with osteoporosis as of 1 November 2018, based on medical records from four health centers of the eastern Valladolid urban health area. This was a random sample of 247 women stratified by health center. Information on osteoporosis risk factors, comorbidities, daily lifestyle habits, and QoL assessed with the EQ-5D was collected. QoL was modeled using sociodemographic variables, lifestyle, and clinical variables. (3) Results: Women with osteoporosis had a positive perception of their health (EQ-5D% VAS 64.9 ± 18.31). High EQ-5D QoL scores were obtained on the dimensions of mobility: 51.6% [95% CI (44.2%, 58.9%)]; self-care: 75.3% [95% CI (68.5%, 81.2%)]; activities of daily living: 71.4% [95% CI (64.4%, 77.6%)]; pain/discomfort: 25.8% [95% CI (19.8%, 32.6%)]; and anxiety/depression: 53.8% [95% CI (46.4%, 61.0%)]. (4) Conclusions: The QoL of the study group was no worse than the QoL of the general population, except for pain/discomfort and anxiety/depression. Age, highest educational level reached, inflammatory diseases, physical activity, and insomnia were independent predictors of QoL in women with osteoporosis.

## 1. Introduction

The population of Spain has one of the highest life expectancies at birth in Europe: 85.7 years for women and 80.4 years for men. The population of older people has become progressively more feminine (32% more women). The population pyramid is now inverted, with people over 65 years of age now representing 19.3% of the population of Spain, and people over 80 years of age, 6.1% [1,2].

Osteoporosis is a systemic skeletal disease whose prevalence is greater in women (4:1 compared to men). Osteoporosis is characterized by a decrease in bone density and strength, originating fragility and fractures over time. It generates significant economic and social costs, increased dependency, and diminished quality of life [3,4,5,6].

The incidence of hospital morbidity due to osteoporosis in the population over 65 years of age in Spain was 57.1% in 2017, a percentage that continues to rise. Osteoporotic fractures originate higher mortality, especially in people over 75 years of age, with important functional consequences and a diminished quality of life in this population [7,8]. The mean hospital stay after a hip fracture due to fragility is about 9.8 days in Spain, and 76% of the cases involve women. Osteo-protective treatments are not widely used to prevent fractures, although the use of calcium and vitamin D increases in post-fracture patients [9].

An assessment of the quality of life of older adults implies evaluating their health status and relationship with their environment: 45.4% of the elderly perceive their health as good or very good, but women have a less favorable perception of their health than men (40.0%, vs. 52.3%) [10,11].

Population aging has increased the incidence of long-term disability and chronic diseases, as well as inequalities related to lifestyle. The life expectancy of women is less healthy in women than in men; women receive less economic benefits due to the situation of the labor market, maternity, and loss of employment, and, in turn, they are the main recipients of aid for autonomy and dependency, remote care, and residential care services [2].

Health and economic policies should promote active aging, and optimizing opportunities for health, participation, and security as a way to improve the quality of life of people as they age [1].

Diverse studies exist on quality of life and osteoporosis, and osteoporosis guidelines to standardize action protocols for diagnosed patients, but there are not enough studies addressing the relationship between risk factors for osteoporosis and quality of life [12,13,14,15].

Numerous instruments for measuring quality of life are known, including the EuroQol-5 Dimension questionnaire (EQ-5D), which is easy to administer in a cross-sectional study, allowing the self-assessment of health status at the time of implementation; the EQ-5D was used in the National Health Survey in Spain (ENSE 2011) in 2011, which was used for purposes of comparison in this study [11,16].

The objective of the study is to assess the quality of life in women over 65 years of age diagnosed with osteoporosis in comparison with the quality of life in the general population, as well as to identify the risk factors associated with osteoporosis that have the greatest impact on quality of life.

## 2. Materials and Methods

An observational, descriptive, cross-sectional study was carried out using quantitative tools to assess the quality of life of women over 65 years of age diagnosed with osteoporosis in the east Valladolid urban health area.

### 2.1. Population and Sample

We chose four urban health centers in the eastern area of Valladolid (La Victoria, Centro-Gamazo, Circunvalación, and Barrio España) that, in the opinion of the researchers, would serve to represent the sociodemographic variety in that area. We identified 704 women in these four centers with a diagnosis of osteoporosis in the computerized medical record (Medoracyl) as of 1 November 2018. A random sample of 247 women stratified by health center and proportionally allocated was obtained: 182 women completed the survey, 17 had severe cognitive impairment, 15 were untraceable, 13 refused to participate in the study, 8 surveys had data coding errors, and 12 were not completed due to work overload.

### 2.2. Inclusion Criteria

Women over 65 years of age diagnosed with osteoporosis who did not have severe cognitive impairment and who voluntarily agreed to participate in the study and gave informed consent were enrolled. 

### 2.3. Variables and Risk Factors

A personal interview was carried out with the women in the study group, which addressed risk factors for osteoporosis and sociodemographic variables. The EQ-5D health questionnaire assesses quality of life through 5 dimensions: mobility, self-care, daily living activities, pain/discomfort, and anxiety/depression, each of which is scored from 1 to 3, where 1 indicates HIGH QoL, 2 MEDIUM QoL, and 3 LOW QoL for each dimension. The numerical result obtained from the combination of responses yielded an objective measure of the quality of life, called the EQ-5D health status, and a complementary value, called the Severity Index. The EQ-5D score was complemented by a visual analog scale (VAS), which each participant used to assess their subjective health status at the time of the survey. The Barthel Index for Activities of Daily Living, Lawton and Brody Instrumental Activities of Daily Living Index, and Folstein Mini-Mental State Examination were also included if they were recorded in the medical history. Healthy lifestyle habits such as exposure to sun, calcium intake, physical activity, dyslipidemia, and insomnia were considered, and measured using a Likert-type scale, which in some cases were summarized as the levels: low, medium, and high; or low and high. For alcohol consumption, physical activity, dyslipidemia, and insomnia, the results were grouped as: 0 points low; 1 to 2 points moderate; 3 to 4 points high. For sun exposure and calcium intake, it was determined that 0 to 2 points were classified as low, and 3 to 4 points were considered high [17]. 

### 2.4. Statistical Analysis

The qualitative variables were summarized as frequency and percentage (fr and %) and the numerical variables as mean and standard deviation (M ± SD). The 95% confidence intervals (95% CI) were obtained for the population percentages and means. The percentages obtained on the 5 dimensions of the EQ-5D were compared with those of the general population obtained in the ENSE 2011. Weighted estimates were obtained from the subsample of women from the ENSE 2011; the weights were based on the age distribution observed in our sample with the aim of ensuring comparability with our study. A test of equality for percentages was used to compare the equality of the frequencies on the EQ-5D dimensions between the women in the study group and those of the general population of the ENSE 2011. The EQ-5D health status obtained from the values of the 5 dimensions was categorized using quartiles. Using multinomial logistic regression models, the relationship between this categorized version of the EQ-5D health status and each explanatory variable was studied, removing the effect of age. Subsequently, a multinomial logistic regression model was estimated to predict values for the aforementioned categorized version of the EQ-5D health status as a function of the explanatory variables. The variables selected after applying a protocol to identify relevant variables, such as that described in Hosmer et al. were included in this model [18]. Based on this estimated model, a score was obtained to predict the quality of life, which is only a function of the explanatory variables. By categorizing this estimated score using quartiles, 4 groups of predicted quality of life were defined. Sensitivity and specificity values were calculated to evaluate the predictive capacity of the classification rules based on the adjusted model. *p* values of less than 0.05 were considered statistically significant. The statistical analyses were completed with R Analytics Software v4.0.

## 3. Results

The means and frequencies of the variables studied, risk factors for osteoporosis, and lifestyle and nutrition habits are shown in Table 1. 

The EQ-5D survey results are summarized in Table 2. The EQ-5D health status score resulting from the objective assessment of quality of life using the five dimensions of the EQ-5D questionnaire was 66.7 ± 24.7 and the self-assessed VAS quality of life was 64.9 ± 18.31 on a scale from 0 to 100. The difference between the objective and subjective assessments was not statistically significant [*p* = 0.15, 95% CI (−4.3, 0.7)]. 

The results of comparing the EQ-5D dimensions of the study group and those of the women from the general population (ENSE 2011) are shown in Figure 1. The frequencies for the five dimensions were: mobility, self-care, daily living activities, pain/discomfort, and anxiety/depression.

The percentage of women with osteoporosis who rated their quality of life as good in terms of the dimensions of mobility, self-care, and activities of daily life was not significantly lower than that observed in women from the general population (ENSE 2011). When making this comparison, women with severe impairment from the general population described in the ENSE 2011 cannot be eliminated because this information was not collected in the survey that year. However, even assuming a scenario in which all the women with severe cognitive impairment eliminated from our sample were outside the highest level of quality of life on these three dimensions, no statistical significance would appear in the previous comparisons. In the pain/discomfort and anxiety/depression dimensions, the percentage of women with osteoporosis with a good quality of life was at least 8.5% lower than the corresponding percentage in the general population (95% confidence). If this worst-case scenario of the women with severe cognitive impairment was eliminated from our study, the percentage of women with a good quality of life would be at least 12% higher among the women in the general population (95% confidence).

The relationship between the EQ-5D health status, or objective assessment of the quality of life of the elements of the sample, and the explanatory variables after removing the effect of age is shown in Table 3.

Age, educational level achieved, presence of inflammatory diseases, physical activity, and insomnia are independent predictors of the quality of life in women over 65 years of age. Classification rules based on this score had a sensitivity/specificity for identifying low EQ-5D health status scores of 68.1%/77.9% and 73.6%/70.9%.

The relationship between the VAS score and the groups derived from categorizing the predicted quality of life score by quartiles is shown in Figure 2.

## 4. Discussion

Older women are often users of primary care services; they live longer than men; they receive fewer economic benefits; and inequalities related to lifestyle factors tend to increase among women over the years [1].

The purpose of this study was to know the quality of life of women over 65 years of age diagnosed with osteoporosis (which has a higher prevalence in women), and to identify the factors that most affect the assessments that people make of their own health and well-being in order to improve them.

The quality of life of women over 65 years of age decreases with the presence of comorbidities such as chronic diseases, increasing their level of dependency. The presence of inflammatory diseases, such as arthritis or osteoarthritis, is an independent predictor of quality of life [13].

The study of quality of life encompasses two aspects: one subjective and the other objective. No statistically significant difference was found between the two in this study. The quality of life of the women studied was no worse than in the general population, except for the pain/discomfort and anxiety/depression dimensions [12].

Some risk factors for osteoporosis that appeared to be significantly related to quality of life, such as age, the presence of inflammatory or kidney diseases, and treatment with corticosteroids, have also been noted in articles and management guidelines on osteoporosis [7,13,14,15,19,20].

Healthy lifestyle habits, such as a balanced diet, regular physical activity, and avoiding a sedentary lifestyle, have been identified as protective factors in relation to osteoporosis and bone fragility, and are associated with a better quality of life [21,22]. In the survey conducted, calcium intake, the only dietary detail addressed, did not appear to be related with a better quality of life.

Adequate nocturnal rest has not typically been related to osteoporosis, although it has been related to optimal quality of life, and was also identified in the study [23,24].

The higher the level of education attained, the better the perceived quality of life is among the elderly; however, the Spain Health System Report in 2018 indicates that its influence has decreased in the last 10 years [25].

Rules based on osteoporosis risk factors can be defined to predict quality of life. Furthermore, if these risk factors are controlled, the effects of osteoporosis will be lessened. It would be advisable to treat inflammatory pathologies that cause pain, as well as anxious-depressive symptoms in these women in order to improve their quality of life. To this end, it is advisable to apply pharmacological and non-pharmacological programs; the latter is based on nursing care and promotion of the active patient [26].

### Limitations

The study was not completed with all the women in the sample because only 79% of the selected women responded. As the ENSE survey did not consider severe cognitive impairment until 2017, when the EQ-5D was not used, we could not eliminate the women with cognitive impairment for comparison with the general population.

## 5. Conclusions

Except for having pain and anxiety or depression, the quality of life of osteoporotic women was no worse than that of the general population.

The results of the study confirm that age, educational level achieved, presence of inflammatory diseases, physical activity, and insomnia are five multivariate predictors of quality of life.

## Figures and Tables

**Figure 1 ijerph-19-05745-f001:**
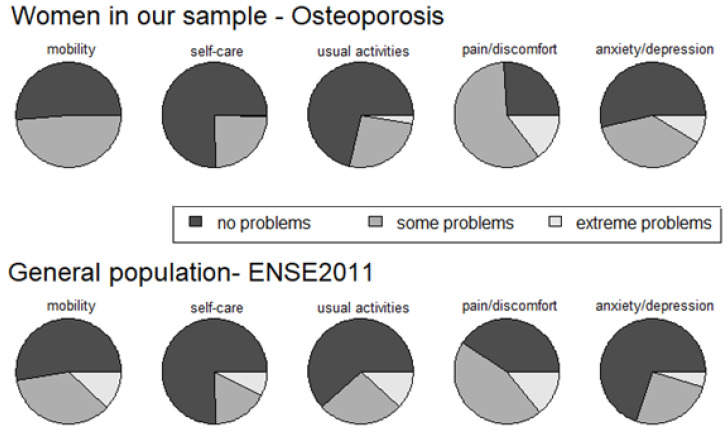
Comparison of the EQ-5D dimensions of the women with osteoporosis in the study group with the ENSE 2011 women.

**Figure 2 ijerph-19-05745-f002:**
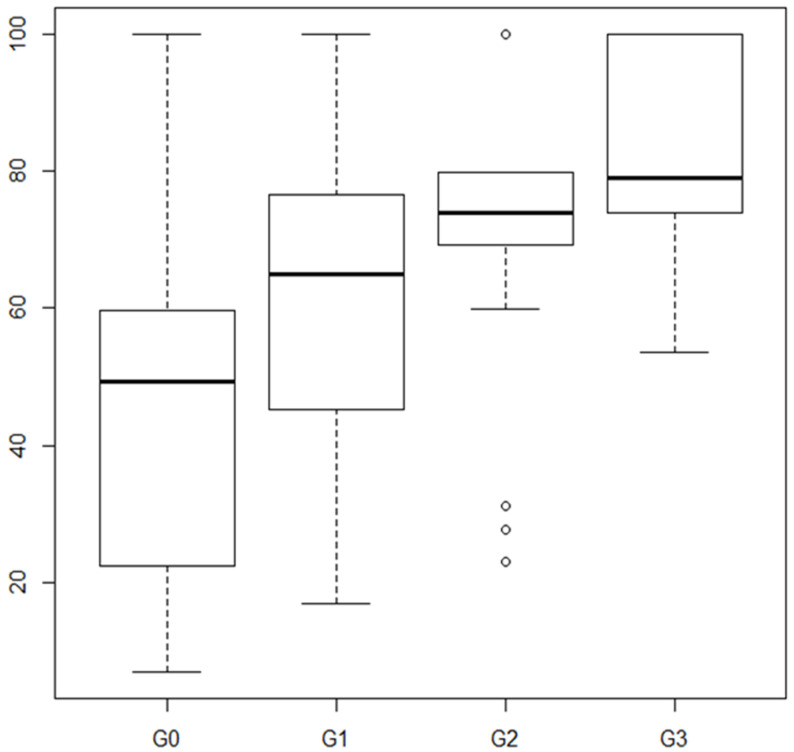
Relationship between quality of life (VAS) and the groups resulting from categorizing the score obtained by the estimated predictive model.

**Table 1 ijerph-19-05745-t001:** Distribution of risk factors for osteoporosis, lifestyle habits, and nutrition.

Variables	*n*	Mean ± SD or Freq (%)	95% CI
Age	182	75.08 ± 6.97	74.06–76.1
Age of Menarche	179	13.28 ± 1.57	13.05–13.51
Age of Menopause	175	49.34 ± 4.81	48.62–50.06
No. of Children	177	2.28 ± 1.37	2.07–2.48
Partner	182	109 (59.9%)	52.5–66.8
Secondary/University Studies	179	48 (27.3%)	21.2–34.3
BMI	166	26.62 ± 4.81	25.88–27.35
Smoker	181	18 (9.9%)	6.4–15.2
Alcohol-Mod	182	12 (6.6%)	3.8–11.1
Alcohol-Heavy	182	7 (3.8%)	1.9–7.7
Physical Activity-Mod	182	80 (44%)	36.9–51.2
Physical Activity-Heavy	182	86 (47.3%)	40.1–54.5
Sun Exposure-Low	181	76 (42%)	35.4–49.3
Sun Exposure-High	181	83 (45.9%)	38.8–53.1
Calcium Intake-Low	179	31 (18.9%)	12.5–23.5
Calcium Intake-High	179	133 (81.1%)	74.4–87.8
Barthel Index	168	92.86 ± 14.44	90.66–95.06
Lawton Index	107	7.1 ± 2.32	6.66–7.55
Mini-Mental State Examination	131	26.66 ± 4.69	25.85–27.47
Corticoids	182	35 (19.2%)	13.8–25.7
Thyroxin	182	32 (17.6%)	12.5–23.8
Antacids	182	56 (30.8%)	24.3–37.9
Furosemide	182	15 (8.2%)	4.7–13
Previous Fractures	182	72 (39.8%)	32.8–47.2
History Osteoporosis	182	71 (40.1%)	33–47.7
Diabetes	182	24 (13.2%)	8.9–18.8
Hypothyroidism	182	41 (22.5%)	16.9–29.3
Hyperthyroidism	182	5 (2.7%)	1.1–6.1
Hyperparathyroidism	182	1 (0.5%)	0–2.8
Renal Disease	182	12 (6.6%)	3.7–11.1
Inflammatory Disease	182	68 (37.6%)	30.6–45
Dyslipidemia-Mod	178	37 (20.8%)	15.5–27.3
Dyslipidemia-High	178	79 (44.4%)	37.3–51.7
Insomnia-Mod	181	63 (34.8%)	28.3–42
Insomnia-High	181	81 (44.8%)	37.7–52

The quantitative variables are expressed as means and standard deviation (SD) and the qualitative values as frequency and percentages (%). There is a 95% confidence interval.

**Table 2 ijerph-19-05745-t002:** EQ-5D. Results.

EQ-5D	N	Mean ± SD or Freq (%)	95% CI
Mobility-Med CV	182	88 (48.4%)	41.1–55.8
Mobility-Low CV	182	0 (0%)	0–2.1
Self-Care-Med CV	182	44 (24.2%)	18.2–30.9
Self-Care-Low CV	182	1 (0.5%)	0–2.8
Usual Activities-Med CV	182	47 (25.8%)	19.8–32.6
Usual Activities-Low CV	182	5 (2.7%)	1.1–6.1
Pain-Med CV	182	108 (59.3%)	51.9–66.3
Pain-Low CV	182	27 (14.8%)	10.3–20.7
Anx/Depress-Med CV	182	68 (37.4%)	30.4–44.8
Anx/Depress-Low CV	182	16 (8.8%)	5.3–13.8
VAS Score	182	64.9 ± 18.31	62.22–67.58
EQ-5D Health Status	182	66.7 ± 24.7	63.1–70.3
Severity Index	182	68.4 ± 31.1	63.9–73.0

The quantitative variables are expressed as means and standard deviation (SD) and the qualitative values as frequency and percentages (%). There is a 95% confidence interval.

**Table 3 ijerph-19-05745-t003:** Relationship between VAS and the explanatory variables. In the dichotomous variables, 0 is NO and 1 is YES. The values 0, 1, 2, and 3 for age, menarche, menopause, and BMI correspond to the respective values for the first, second, third, and fourth quartiles. For the number of children, 0 corresponds to no children, 1 to one child, and 2 to two or more children. For the variables physical activity, sun exposure, and calcium intake, the scale from 0 to 4 corresponds to a Likert-type scale where 0 indicates absence and 4 indicates maximum adherence/exposure to the associated factor.

	0/NO	1/YES	2	3	4	Univariate *p*-Value (–Age)	Multivariate *p*-Value
Age	73.6	75.8	66.9	54.3		<0.001 ***	0.005 **
Menarche	58.5	71.3	67.1	65.7		0.895	
Menopause	65.5	65.3	67.7	67.8		0.688	
No. Children	63.4	71.2	64.6			0.335	
Partner	62.8	69.3				0.851	
Education	59.1	64.1	76.0			0.001 **	0.002 **
BMI	68.9	67.0	69.2	62.8		0.03 *	
Tobacco	66.7	67.9	67.9	57.1		0.286	
Alcohol	66.1	74.4	65.9	76.0		0.652	
Physical Act	44.8	52.9	69.9	72.9	75.4	<0.001 ***	0.004 **
Sun Expos	54.0	65.0	66.5	63.7	74.1	0.009 **	
Ca Intake	68.2	50.0	74.4	64.8	66.6	0.994	
Corticoids	69.5	54.8				0.001 **	
Thyroxin	67.7	62.2				0.155	
Antacids	70.5	58.1				0.011 *	
Furosemide	67.1	62.4				0.931	
Fractures	71.2	59.9				0.011 *	
History Osteopor	69.9	62.0				0.013 *	
Diabetes	67.8	59.5				0.357	
Hypothyroidism	68.5	60.5				0.042 *	
Hyperthyroidism	66.6	70.4				0.587	
Hyperparathy	66.7	70.1				0.852	
Renal Disease	68.8	37.9				<0.001 ***	
Inflammatory Dis	72.3	57.5				<0.001 ***	0.007 **
Dyslipidemia	67.6	71.3	63.4	67.4	63.9	0.53	
Insomnia	78.5	67.4	68.5	58.3	60.9	<0.001 ***	0.002 **

* *p* < 0.05; ** *p* < 0.01; *** *p* < 0.001.

## Data Availability

Not applicable.

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
