# Peer review of "Quality of Life in Women over 65 Years of Age Diagnosed with Osteoporosis"

_ijerph, 2022, doi:10.3390/ijerph19095745_

Round 1

Reviewer 1 Report

This is a good study for women over 65 years of age diagnosed with osteoporosis and in the study demonstrated age, highest educational level reached, inflammatory diseases, physical activity, and insomnia were independent predictors of quality of life in women with osteoporosis. These important findings can be supported to build health and economic policies to help clinical functional maintain and prevent further medical cost.

After English correction the study can be accepted for publication.

Author Response

Dear reviewer,

We very much appreciate your favourable comments on the article. We agree on the benefits that our research can bring to the healthcare system.

Regarding the English translation, we would like to point out that it has been done by a company specialised in scientific translations, "Eurotext", and that we have an accreditation document certifying this. If you consider it necessary to check it, we are at your disposal to provide you with it.

Thank you very much for your review.

The research team.

Reviewer 2 Report

An interesting study, What is missing (for the reviewer): Which factors can be reduced to improve quality of life? Lines 203/204 and 226/227 report effects of "pain-discomfort" and "anxiety-depression": Modern psycho-pharmacology a.o.  offer several ways to reduce these factors; hence it would be helpful to include proposals. - Did the women not chose these possibilities?--- Please also define what is "high" and "low" conc. Ca intake and physical activity - and list  "antacids" and "inflammatory disease"

--

Author Response

Dear reviewer,

We very much appreciate your favourable comments on the article.

Regarding the reduction of risk factors to improve the quality of life of these women, we have specified how to reduce pain and discomfort, as well as anxiety and depression symptomatology that influence women's quality of life. The proposals are based on other articles, as these aspects were not really explored in this study.

In the case of alcohol consumption, physical activity, dyslipidaemia and insomnia, the results were grouped into: o points low; 1 to 2 points moderate; 3 to 4 points high. For sun exposure and calcium intake, 0 to 2 points were classified as low and 3 to 4 points as high.

The decisions on these risk factor classification thresholds were purely on statistical criteria. This description is included in the study to avoid confusion.

The drugs consumed were noted by the nurse who collected the data from the patient's medical records. In the clinical history, the drugs are classified according to the classification of active ingredients established by the Spanish Medicines Agency. It can be consulted on the website: https://www.aemps.gob.es/eu/industria/etiquetado/conduccion/listadosPrincipios/grupo-A.htm

Thank you very much for your review and feedback.

The research team.

Reviewer 3 Report

The article written by Pastor -Robles et al explained, if women (over age of 65) suffering from osteoporosis have an affected quality of life including education, pain/discomfort, anxiety/depression as compared to general population.  The study and contents of the manuscript are not very modern and thriving, however overall study is clear, concise and well written.  The methods are generally appropriate, although clarification of a few details can give more detail understating. The results obtained from the studies have been presented quite well. The conclusions drawn are well supported by the literature. On the basis of these facts, I support the publication of this manuscript.

Author Response

Dear reviewer,

We very much appreciate your favourable comments on the article. The results may not be entirely novel, but we believe that they adequately support other research on this topic and add to the existing scientific literature. Moreover, in healthcare practice this problem remains unresolved, and we believe it is necessary to continue to emphasise it. However, we understand your assessment.

We would like to point out that some aspects of the article have been specified to improve its understanding. We hope that this will improve the article.

Thank you very much for your review and feedback.

The research team.